# Effect of Designating Emergency Medical Centers for Critical Care on Emergency Medical Service Systems during the COVID-19 Pandemic: A Retrospective Observational Study

**DOI:** 10.3390/jcm11040906

**Published:** 2022-02-09

**Authors:** Hang A Park, Sola Kim, Sang Ook Ha, Sangsoo Han, ChoungAh Lee

**Affiliations:** 1Department of Emergency Medicine, Dongtan Sacred Heart Hospital, Hwaseong-si 18450, Korea; hangapark@hallym.or.kr (H.A.P.); solarsolakim@hallym.or.kr (S.K.); 2Department of Emergency Medicine, Hallym University Sacred Heart Hospital, Anyang-si 14068, Korea; mdhso@hallym.or.kr; 3Department of Emergency Medicine, Soonchunhyang University Bucheon Hospital, Bucheon 14584, Korea; brayden0819@daum.net

**Keywords:** COVID-19, emergency medical services, ambulance diversion, prehospital, time factors

## Abstract

During the coronavirus disease 2019 (COVID-19) pandemic, prehospital times were delayed for patients who needed to arrive at the hospital in a timely manner to receive treatment. To address this, in March 2020, the Korean government designated emergency medical centers for critical care (EMC-CC). This study retrospectively analyzed whether this intervention effectively reduced ambulance diversion (AD) and shortened prehospital times using emergency medical service records from 219,763 patients from the Gyeonggi Province, collected between 1 January and 31 December 2020. We included non-traumatic patients aged 18 years or older. We used interrupted time series analysis to investigate the intervention effects on the daily AD rate and compared prehospital times before and after the intervention. Following the intervention, the proportion of patients transported 30–35 km and 50 km or more was 13.8% and 5.7%, respectively, indicating an increased distance compared to before the intervention. Although the change in the AD rate was insignificant, the daily AD rate significantly decreased after the intervention. Prehospital times significantly increased after the intervention in all patients (*p* < 0.001) and by disease group; all prehospital times except for the scene time of cardiac arrest patients increased. In order to achieve optimal treatment times for critically ill patients in a situation that pushes the limits of the medical system, such as the COVID-19 pandemic, even regional distribution of EMC-CC may be necessary, and priority should be given to the allocation of care for patients with mild symptoms.

## 1. Introduction

Since the first report of the coronavirus disease 2019 (COVID-19) in Wuhan, China in December 2019, the number of cases and deaths has increased worldwide, and on 1 March 2020, the World Health Organization (WHO) declared a pandemic [1]. According to a WHO report, there were 299 million cases and 5.5 million deaths worldwide as of January 2022 [2]. In addition to its social and economic consequences, COVID-19 has directly impacted the health care system [3]. In the early stages of the pandemic, the number of emergency department (ED) visits decreased and the hospitalization rate for time-sensitive diseases such as acute coronary syndrome (ACS), acute ischemic stroke, and acute respiratory distress syndrome (ARDS) also decreased [4]. However, as COVID-19 continues to spread and the number of cases continues to rise, the number of emergency medical service (EMS) calls increases as well [5]. The length of stay in the ED has increased significantly during the pandemic compared to pre-pandemic times [6] and hospitalizations for critically ill patients in the ED have doubled [7]. Consequently, issues with the transportation of emergency patients to the ED have been reported in many countries [8,9].

Velasco et al. reported the prehospital times during the COVID-19 pandemic [9]. These delays are critical for patients needing to arrive at the hospital in a timely manner to receive treatment. If the prehospital time is delayed, the treatment outcome or prognosis may be poor [10,11,12]. Therefore, it is necessary to immediately screen patients with time-sensitive diseases on the scene and promptly select and transport them to a hospital that can treat them.

To address this problem, the Korean government has attempted to implement a treatment system for critically ill patients by designating emergency medical centers for critical care (EMC-CC) in each region, establishing a real-time monitoring system for their ability to accept patients, and activating the use of a prehospital severity classification system. In addition, EMC-CC were designated to treat patients with time-sensitive diseases in their local region. Financial support was also provided to the EMC-CC to secure and operate isolation rooms in the ED. In addition, hospitals other than EMC-CC were assigned to treat patients with mild symptoms.

This study used the EMS database to evaluate whether this new intervention has been effective in reducing ambulance diversions and shortening prehospital times for patients.

## 2. Materials and Methods

### 2.1. Study Design and Setting 

This retrospective observational study was conducted using EMS records in the Gyeonggi Province. According to estimates from 2021, the Gyeonggi Province in South Korea has an area of 10,172 km^2^ and a population of 13,500,000. The EMS system of Gyeonggi is government-based and provides a basic to intermediate level EMS from fire agency headquarters. As of 2021, the Gyeonggi EMS system consists of 2 headquarters, 35 fire stations with 263 ambulances, and 1912 paramedics [13]. There are two EMS system control centers, and all emergency calls are processed and dispatched to the EMS teams. Basically, the EMS is a single-tiered system. One ambulance team has two or three crew members, including at least one intermediate-level emergency medical technician (EMT). The team can evaluate a patient’s condition and perform fluid resuscitation and advanced airway management under a physician’s direct medical supervision, depending on the patient’s condition. For patients with out-of-hospital cardiac arrest only, a multi-tiered system is employed. Four to six EMTs perform advanced resuscitation, and if the patient is not resuscitated after continuous efforts or return of spontaneous circulation, the patient is transferred to the hospital.

There are a total of 64 EDs in Gyeonggi, most of which strengthened the screening and preemptive isolation of patients with COVID-19-related symptoms through triage based on the following criteria: (1) body temperature of 37.5 °C or higher; (2) presence of symptoms of upper respiratory tract infection such as cough, sputum, or runny nose; and (3) symptoms of respiratory distress or desaturation [14].

### 2.2. Dataset

This study analyzed the EMS run sheet data collected between 1 January and 31 December 2020. We included patients aged 18 years or older who were transported to the hospital via EMS. Patients who called for the EMS for injuries and patients whose time variables, destination hospital, or prehospital vital signs were unknown were excluded.

### 2.3. Intervention

In March 2020, the government implemented interventions to ensure that patients received all their treatment at an institution in their area of residence without having to be transported to other areas, especially if the local institution could handle the case. In the prehospital stage, the triage system was reinforced and the capacity of general beds and isolation rooms in the EDs of regional medical institutions were monitored in real time to identify available hospitals. In addition, the government designated EMC-CC and financed the expansion and operation of isolation rooms. The EMC-CC have given priority to seriously ill patients with suspected COVID-19 within the region (Figure 1). The patient care process was conducted independently according to the circumstances of each EMC-CC (Table 1). In addition, patients with mild symptoms were referred to other emergency medical institutions within their jurisdictions.

### 2.4. Variables

When EMS providers arrive at a scene, they identify and record the patients’ information, chief complaints, and vital signs [15]. In addition to basic information, time variables such as EMS call time, dispatch time, scene arrival time, scene departure time, hospital arrival time, destination hospital, distance to hospital from the scene, and ambulance diversion (AD) are recorded. For patients with cardiac arrest or suspected heart disease or stroke, EMS providers must conduct and document a detailed evaluation. These EMS data have the same variables and structure and are managed by each EMS headquarter in 17 cities and provinces [16]. 

In this study, patients were classified into the following categories based on their EMS records: cardiac arrest, ACS suspected, acute stroke suspected, mental change, critically ill patients, and COVID-19 suspected. Acute stroke patients were screened on the scene using the Cincinnati prehospital stroke scale [17]. Patients’ level of consciousness was evaluated using the “alert, verbal, pain, unresponsive” (AVPU) scale and indications of pain or unresponsiveness were defined as a mental change [18]. Additionally, if the ratio of the heart rate (HR) to systolic blood pressure (SBP) was greater than 1.0 (shock index (SI) > 1.0) [19], the patient was classified as critically ill. Those with upper respiratory symptoms such as cough, sputum, rhinorrhea, difficulty breathing, or fever were classified as COVID-19 suspected. 

### 2.5. Outcomes 

In this study, EMS transport failure was measured using AD as the primary outcome and prehospital time as the secondary outcome. AD was defined as a case in which an ambulance was redirected because the initial ED was unable to provide care for the patient. Prehospital time consisted of response time, on-scene time, and transport time [20] and was calculated using the time variables recorded in the data. 

### 2.6. Statistical Analysis

The characteristics of the study subjects measured both before and after the intervention are presented. In addition, we analyzed whether there was a difference in the proportion of critically ill patients admitted to the EMC-CC before and after the intervention. Differences in the variables between the two periods were tested using the χ^2^-test, and categorical variables are presented as numbers and percentages. 

Interrupted time series (ITS) analysis using segmented Poisson regression was used to investigate the effects of the intervention on the daily AD rate of total EMS responses and the daily AD rate of COVID-19 suspected patients. A quasi-Poisson regression model was used to address the over-dispersion. The number of patients with confirmed COVID-19 from the previous day was adjusted for in the model. This method was useful for identifying changes in levels and trends after the intervention [21]. A segmented Poisson model can be expressed by an equation using time (T) and intervention (I) as follows: ln(λ) = β0 + β1 (T) + β2 (I) + β3 (T*). In this equation, T* represents the time after the intervention, β1 is the trend before the intervention, β2 is the change in level, and β3 represents the difference between trends before and after the intervention [22].

The prehospital times before and after the intervention are presented using the median and interquartile range (IQR). Differences in the pre-hospital time for each disease group were assessed using the Mann-Whitney U-test. All statistical analyses were performed using R (R Foundation for Statistical Computing, Vienna, Austria) version 4.0.1, and statistical significance was defined as *p* < 0.05.

## 3. Results

The total number of EMS responses during the study period was 404,485, with 219,763 cases included in the analysis (Figure 2). The proportion of patients over 65 years old was 47% before the intervention period, higher than after the intervention, and there was no significant difference according to sex (Table 2). In total, the most frequently reported distance to the final hospital was 5–10 km, followed by 10–15 km (37.3% and 31.7%, respectively), which was the same before and after the intervention. However, after the intervention, 13.8% of patients were transported 30–35 km and 5.7% were transported 50 km or more, indicating an increase in transport distance compared to before the intervention. As for the disease category, the proportion of patients with suspected ACS or SI > 1.0 increased and the proportion of patients with cardiac arrest decreased during the post-intervention period. The AD rate decreased significantly after the intervention. 

Table 3 shows the transport rates to the EMC-CC by disease among all EMS responses before and after the intervention. Among all EMS responses, 37,618 cases (22.2%) were transported to the EMC-CC during the post-intervention period which is significantly higher than the rate of 21.7% before the intervention. In addition, the number of suspected COVID-19 patients who were transported to the EMC-CC before the intervention was 30,460 (21.8%), but after the intervention the proportion increased significantly to 23.8%. However, there was no significant difference in transport rate to the EMC-CC by disease group before and after the intervention, except for patients with mental change who were suspected of having COVID-19. 

Table 4 shows the change in the daily AD rate after intervention based on interrupted segmented analysis. As of 23 March 2020, prior to the intervention, the daily AD rate tended to increase by 0.3%, but this was not significant. The change in the daily AD rate due to the intervention was also found to be insignificant. However, after the intervention, the daily AD rate significantly decreased by 0.6%. The rate of COVID-19 suspected patients increased by 0.7% every day before the intervention and decreased by 1.1% thereafter. The number of confirmed cases of COVID-19 from the previous day was significantly associated with the daily AD rate in both the total patients and those with suspected COVID-19.

The total prehospital time significantly increased after the intervention in all patients (*p* < 0.001); in particular, transport time saw the largest increase after the intervention (Figure 3). When the prehospital time was assessed by the disease group, it was found that all prehospital times, except for the scene time of cardiac arrest patients, increased (Table 5).

## 4. Discussion

The designation of the EMC-CC aimed to provide treatment without delay by prioritizing transport to hospitals where patient treatment was appropriate and promptly performed. This intervention reduced AD but did not reduce the prehospital time.

In this study, the AD rate and prehospital time were used as indicators of intervention success. In general, AD is one way to reduce the burden on an ED by diverting patients transported by ambulance to a nearby hospital [23]. At the same time, however, AD causes overcrowding and increases the mortality rates in other EDs in the same service area [24]. This was addressed in Stockholm, Sweden [25] by transporting patients with respiratory symptoms to one hospital, installing intensive care units to reduce the burden on neighboring hospitals, and preventing AD during the COVID-19 outbreak. AD is also associated with a delay in prehospital time [23]. Although this study found that AD rates decreased after the intervention, an increase in the prehospital time was identified during the COVID-19 pandemic. Ageta et al. showed that prehospital time significantly increased during COVID-19 when all patients transported by EMS were analyzed, excluding interfacility transport [26]. Katayama et al. also reported an increase in the number of patients with increased transport times who were hospitalized for acute diseases [8]. Prolonged prehospital times can result in missing the golden hour in patients with the time-sensitive disease.

Time-sensitive disease groups were defined as those with out-of-hospital cardiac arrest, ACS, or acute stroke. We also tried to classify the critical cases, but that was difficult based on a prehospital evaluation [27]. We, therefore, used SI and mental change as indicators for critically ill patients. Studies have shown that SI can be a tool for early recognition and evaluation of critical illness in prehospital settings [28,29] and SI ≥ 1.0 indicates worsening of hemodynamic status and shock [19]. As reported by other studies [30,31,32], prehospital times, including response time and transportation time in all time-sensitive disease categories, continues to increase. 

Several factors may contribute to the failure of the intervention to achieve its objectives. First, the intervention focused on the prehospital phase. The model for evaluating the effectiveness of the AD strategy on ED overcrowding follows the conceptual input-throughput-output framework proposed by Asplin et al. [33]. These results suggest that the ED-based patient flow that adds a fast track unit or improves ED laboratory turnaround time has an effect on reducing AD [34]. In addition, it is difficult to fundamentally solve the overcrowding of the ED unless there are improvements in patient flow that enable increased output and timely admission for inpatient care [25]. Ambulances were directly sent to designated hospitals that provide critical care and additional isolation rooms were secured in the EMC-CC through financial support. However, there was no way to quickly test for COVID-19, and the process related to in-hospital admission did not improve, limiting the ability to accommodate critically ill patients.

The second factor is the distributional bias of the EMC-CC. As shown in Figure 1, the EMC-CCs are not located in central areas, but are biased; therefore, a deviation in the transport distance within an area is unavoidable. Because EMC-CC were functionally converted from the existing EMC, it was not possible to cover areas lacking existing medical resources. As a result, the transport distance to the final hospital where critically ill patients were treated increased, which also increased the transport time. These results are consistent with a previous study that showed that the centralization of facilities increases the transport time by increasing the transport distance of the ambulance [35]. In addition, the longer the transport distance, the longer the turnaround time of ambulances which delays the response time for other patients [36].

The final factor is the failure to disperse patients with mild symptoms. The purpose of the EMC-CC designation is to ensure that critically ill patients with suspected COVID-19 have priority access to appropriate treatment. Our results showed that the proportion of total patients transported to the EMC-CC increased; however, there was no difference in the proportion of critically ill patients transported to the EMC-CC before and after the intervention. These results indicate an increase in the transport rate of patients with mild symptoms and suspected COVID-19. In Stockholm, one ED could not accommodate all patients with respiratory disease, so some patients were given priority while some were dispersed to other hospitals [25]. Therefore, if patients with mild symptoms who are suspected to have COVID-19 are not more actively dispersed to other hospitals, the window for the optimal treatment time may be missed due to EMC-CC overcrowding. 

### Limitations 

This study had several limitations. First, patients were evaluated based on EMS records. In other words, classifications may not have been objective because they were based on the patients’ chief complaints and evaluations by EMS providers in the field. We attempted to address this by using EMS screening protocols for ACS and stroke. Second, since this study was based on EMS run sheet data, there was no clinical outcome information following treatment. Therefore, it is not known whether the transported patients received appropriate treatment at the hospital. However, a stipulation of EMC-CC designation is that treatment facility standards must be met, so it is presumed that the final treatment was provided. Third, because information on patients who visited the ED without EMS was not included in the analysis, it was not possible to determine the number of critically ill patients who did not use EMS. This may include patients who arrived at the hospital using other means of transportation, although it is known that EMS is more likely to be used for critically ill patients [37]. Finally, the data used in this study were before the spread of the Omicron variant. If the variant progresses to a more transmissible, but lower severity pattern, the focus of treatment may need to be changed to be clinic-based or home care rather than using EMC-CC.

## 5. Conclusions

In order to achieve optimal treatment times for critically ill patients in a situation that pushes the limits of the medical system, such as the COVID-19 pandemic, even regional distribution of EMC-CC may be necessary to improve intervention and priority should be given to the allocation of care for patients with mild symptoms. Moreover, further studies are needed to identify factors for improving interventions.

## Figures and Tables

**Figure 1 jcm-11-00906-f001:**
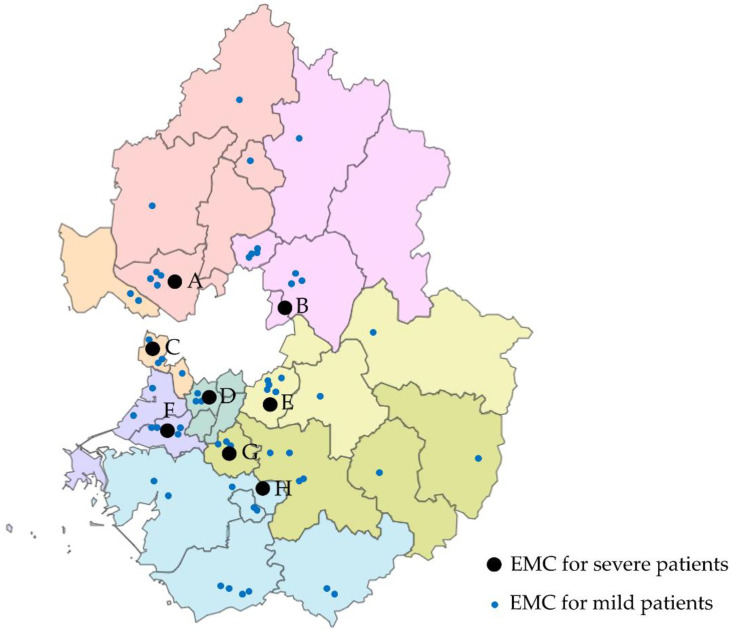
Distribution of emergency medical centers for critical care (EMC-CC) in Gyeonggi province.

**Figure 2 jcm-11-00906-f002:**
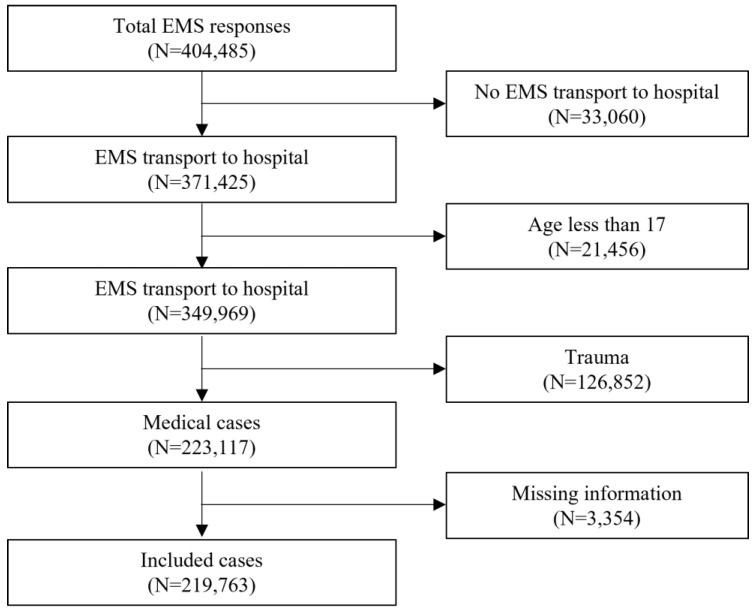
Flow chart of patient inclusion criteria. EMS: emergency medical service.

**Figure 3 jcm-11-00906-f003:**
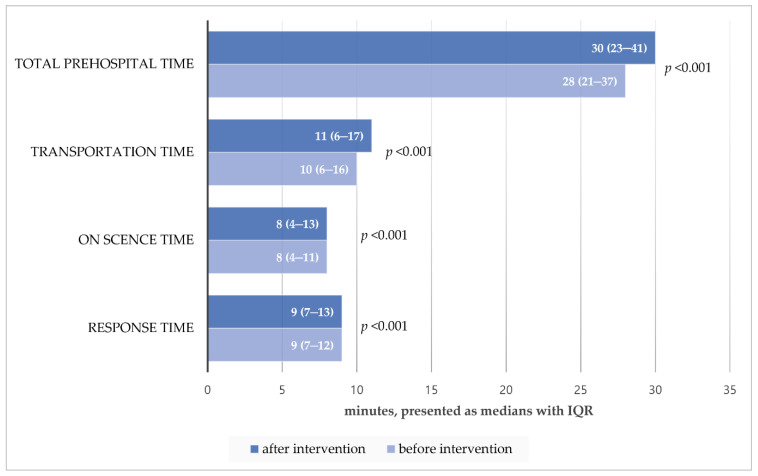
EMS prehospital times by intervention among all EMS responses. Abbreviations: EMS, emergency medical service; IQR, interquartile range.

**Table 1 jcm-11-00906-t001:** Change of facilities and operation before and after EMC-CC designation.

EMC-CC	Before		After		
	Total Number of Beds	Number of Isolation Rooms	Total Number of Beds	Number of Isolation Rooms	Others
A	28	7	26	10	None
B	22	3	22	5	None
C	29	5	29	9	None
D	42	7	39	11	None
E	36	5	41	10	None
F	22	6	22	6	Cohort isolation ward: 8 beds for critically ill patients Fever clinic (6 p.m.–2 a.m.)
G	36	8	36	8	Cohort isolation ward: 12 beds for general patients, 4 beds for critically ill patients Fever clinic
H	46	3	38	5	Cohort isolation ward: 7 beds for general patients (April 2020–November 2020)

Before and after intervention was classified as before or after 23 March 2020. Abbreviations: EMC-CC, emergency medical center for critical care.

**Table 2 jcm-11-00906-t002:** Baseline characteristics of the study subjects before and after the intervention.

Variables	Total		Before		After		*p*
	N	%	N	%	N	%	
Total	219,763		50,049		169,714		
Age							<0.001
18–64 years	119,352	54.3	26,519	53.0	92,833	54.7	
65 years or older	100,411	45.7	23,530	47.0	76,881	45.3	
Sex							0.09
female	108,200	49.2	24,807	49.6	83,393	49.1	
male	111,563	50.8	25,242	50.4	86,321	50.9	
Time							0.03
00:00–05:59	37,204	16.9	8659	17.3	28,545	16.8	
06:00–11:59	66,845	30.4	15,028	30.0	51,817	30.5	
12:00–17:59	61,209	27.9	13,964	27.9	47,245	27.8	
18:00–23:59	54,505	24.8	12,398	24.8	42,107	24.8	
Day of week							<0.001
weekend	59,910	27.3	13,961	27.9	45,949	27.1	
weekday	159,853	72.7	36,088	72.1	123,765	72.9	
Distance from scene to hospital							<0.001
<5 km	1354	0.6	396	0.8	958	0.6	
5–10 km	82,008	37.3	19,872	39.7	62,136	36.6	
10–15 km	69,642	31.7	16,138	32.2	53,504	31.5	
15–30 km	25,507	11.6	5676	11.3	19,831	11.7	
30–50 km	29,318	13.3	5817	11.6	23,501	13.8	
≥50 km	11,831	5.4	2127	4.2	9704	5.7	
Unknown	103	0.05	23	0.05	80	0.05	
Disease Category							
Cardiac arrest	5502	2.5	1341	2.7	4161	2.5	0.004
Mental change	15,858	7.2	3686	7.4	12,172	7.2	0.14
Stroke suspected	4247	1.9	948	1.9	3299	1.9	0.48
ACS suspected	9941	4.5	2156	4.3	7785	4.6	0.01
SI > 1.0	17,187	7.8	3688	7.4	13,499	8	<0.001
Ambulance diversion	1424	0.6	370	0.7	1054	0.6	0.004

Before and after intervention was classified as before or after 23 March 2020. Abbreviations: N, number; ACS, acute coronary syndrome; SI, shock index.

**Table 3 jcm-11-00906-t003:** Rates of EMS transport to EMC-CC by disease group, before and after intervention periods.

	Before			After			*p*
	N of EMS Responses	N of EMS Responses Transported to EMC-CC	%	N of EMS Responses	N of EMS Responses Transported to EMC-CC	%	
Total EMS responses	50,049	10,841	21.7	169,714	37,618	22.2	0.02
COVID-19 suspected	30,460	6639	21.8	91,024	21,661	23.8	<0.001
Cardiac arrest							
COVID-19 non-suspected	46	7	15.2	258	50	19.4	0.51
COVID-19 suspected	1295	303	23.4	3903	990	25.4	0.16
Mental change							
COVID-19 non-suspected	1083	235	21.7	3968	864	21.8	0.96
COVID-19 suspected	2603	643	24.7	8204	2206	26.9	0.03
Stroke suspected							
COVID-19 non-suspected	655	199	30.4	2428	780	32.1	0.40
COVID-19 suspected	293	91	31.1	871	288	33.1	0.53
ACS suspected							
COVID-19 non-suspected	1134	332	29.3	4520	1274	28.2	0.47
COVID-19 suspected	1022	322	31.5	3265	1018	31.2	0.84
SI > 1.0							
COVID-19 non-suspected	1059	263	24.8	4370	1044	23.9	0.52
COVID-19 suspected	2629	737	28.0	9129	2692	29.5	0.15

Before and after intervention was classified as before or after 23 March 2020. Abbreviations: EMS, emergency medical service; EMC-CC, emergency medical center for critical care; N, number; COVID-19, coronavirus disease 2019; ACS, acute coronary syndrome; SI, shock index.

**Table 4 jcm-11-00906-t004:** Results of segmented Poisson regression analysis of an interrupted time series exploring the effect of the intervention on the daily ambulance diversion rate.

	Coefficient	Rate	95% CI	*P*
Total EMS responses				
β_0_: intercept	−5.077	0.006	0.005–0008	<0.001
β_1_: slope before intervention	0.003	1.003	0.999–1.008	0.16
β_2_: level change after intervention	0.007	1.007	0.768–1.320	0.96
β_3_: slope change after intervention	−0.006	0.994	0.990–0.999	0.01
β_4_: yesterday’s number of confirmed COVID-19 cases	0.0004	1.0004	1.0001–1.007	0.02
COVID-19 suspected cases				
β_0_: intercept	−5.405	0.004	0.003–0.006	<0.001
β_1_: slope before intervention	0.007	1.007	1.001–1.014	0.02
β_2_: level change after intervention	−0.082	0.921	0.635–1.337	0.67
β_3_: slope change after intervention	−0.011	0.989	0.982–0.995	<0.001
β_4_: yesterday’s number of confirmed COVID-19 cases	0.001	1.001	1.0002–1.001	0.01

Before and after intervention was classified as before or after 23 March 2020. The intercept (β_0_) refers to the baseline level before the intervention. β_1_ refers to the slope of daily ambulance diversion rates before the intervention, β_2_ refers to the immediate change in the level of daily ambulance diversion rates after the intervention, and β_3_ refers to the difference between slopes of daily ambulance diversion rates before and after the intervention. Abbreviations: CI, confidence interval; COVID-19, coronavirus disease 2019.

**Table 5 jcm-11-00906-t005:** EMS pre-hospital times before and after the intervention by disease group.

	Total		Before		After		*p*
	Median	IQR	Median	IQR	Median	IQR	
COVID-19 suspected							
Response time	9	(7–12)	9	(7–12)	9	(7–13)	<0.001
On scene time	8	(4–13)	8	(4–11)	9	(5–13)	<0.001
Transportation time	9	(5–15)	9	(5–14)	9	(5–16)	<0.001
Total prehospital time	29	(21–39)	26	(20–35)	29	(22–40)	<0.001
Cardiac arrest							
Response time	9	(7–11)	8	(6–10)	9	(7–11)	<0.001
On scene time	16	(12–20)	15	(12–20)	16	(12–20)	0.14
Transportation time	7	(5–12)	7	(4–11)	8	(5–12)	<0.001
Total prehospital Time	33	(27–41)	32	(26–39)	33	(28–42)	<0.001
ACS suspected							
Response Time	9	(7–12)	9	(6–11)	9	(7–12)	<0.001
On Scene Time	6	(3–9)	5	(2–9)	6	(3–10)	<0.001
Transportation Time	12	(7–19)	11	(7–18)	12	(7–19)	0.03
Total prehospital time	28	(22–38)	27	(21–36)	29	(22–39)	<0.001
Stroke suspected							
Response time	9	(7–12)	9	(7–12)	9	(7–13)	<0.001
On scene time	9	(6–13)	8	(6–11)	9	(6–13)	<0.001
Transportation time	13	(7–20)	12	(7–19)	13	(8–20)	0.02
Total prehospital time	33	(25–43)	31	(23–41)	34	(26–44)	<0.001
SI > 1.0							
Response time	9	(7–13)	9	(7–12)	10	(7–13)	<0.001
On scene time	10	(7–15)	9	(6–13)	11	(7–16)	<0.001
Transportation time	12	(7–20)	11	(7–18)	13	(8–21)	<0.001
Total prehospital time	35	(26–46)	31	(24–41)	36	(27–48)	<0.001
Mental change							
Response time	9	(7–12)	8	(6–11)	9	(7–12)	<0.001
On scene time	14	(10–18)	13	(9–17)	14	(10–19)	<0.001
Transportation time	10	(6–17)	9	(5–14)	10	(6–17)	<0.001
Total prehospital time	35	(27–44)	32	(25–40)	35	(28–46)	<0.001

Before and after intervention was classified as before or after 23 March 2020. Abbreviations: EMS, emergency medical service; ACS, acute coronary syndrome; SI, shock index.

## Data Availability

The data used to support the findings of this study are available from the corresponding author upon request.

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
