# Peer review of "Effect of Designating Emergency Medical Centers for Critical Care on Emergency Medical Service Systems during the COVID-19 Pandemic: A Retrospective Observational Study"

_jcm, 2022, doi:10.3390/jcm11040906_

Round 1

Reviewer 1 Report

I recommend that the authors have an independent biostatistician review their statistical methods and results, especially those in Table 5. The unbalanced design identified by comparison groups as N=50,049 for before and N=169,714 for after (in Table 2) after raises some questions. The same is true of the N's displayed in Table 3 before and after the intervention.

The biostatistician could prepare a validation letter that is sent to the Editor.

This statement in the Conclusion (line 283), "...the regional distribution of specialized medical institutions should be equalized..." is somewhat vague, and perhaps the biostatistician can help the authors refine the statement, make it more explicit, and/or clarify what the authors are saying in terms of concrete claims based on sound P values in their data analysis.

Generally, the paper is well written and useful for future planning, but should include some projection for what now, particularly with the Omicron variant, has become an endemic challenge to emergency medical systems worldwide.

Author Response

We sincerely appreciate the careful review of our manuscript and the helpful suggestions provided, which have considerably contributed to improving our manuscript. The manuscript has been revised as suggested.

I recommend that the authors have an independent biostatistician review their statistical methods and results, especially those in Table 5. The unbalanced design identified by comparison groups as N=50,049 for before and N=169,714 for after (in Table 2) after raises some questions. The same is true of the N's displayed in Table 3 before and after the intervention.

The biostatistician could prepare a validation letter that is sent to the Editor.

→ Thank you for your valuable suggestions. We requested for an independent biostatistician’s review of our statistical analysis, and were told that, although there was an imbalance in the number of cases before and after intervention, the statistical methods and results could be maintained. A biostatistician’s evaluation report is also submitted.

This statement in the Conclusion (line 283), "...the regional distribution of specialized medical institutions should be equalized..." is somewhat vague, and perhaps the biostatistician can help the authors refine the statement, make it more explicit, and/or clarify what the authors are saying in terms of concrete claims based on sound P values in their data analysis.

→ We totally agree that the conclusions should be clear and explicit. Therefore, the expression "...the regional distribution of specialized medical institutions should be equalized...", was amended to "…...an even regional distribution of EMC-CC may be necessary to improve an intervention, ….” Additionally, we stated in the conclusion that further research is needed. (Conclusion section, page 12, lines 292-295)

Generally, the paper is well written and useful for future planning, but should include some projection for what now, particularly with the Omicron variant, has become an endemic challenge to emergency medical systems worldwide

→ Thank you for your important suggestion. The data used in this study are those before the spread of the Omicron variant. The need of a change in the response strategy according to the variants’ pattern was additionally described in the limitation section. (Page 12, lines 286-288)

Reviewer 2 Report

Many compliments to the authors of this paper for their analysis
and attention to pre-hospital times during the COVID 19 pandemic
that has put the health system under pressure all over the world.
Nice idea. Well written and well illustrated work.

I ask the authors to clarify which were the figures (medical and paramedical) that they managed in the pre-hospital phase, both on site during the intervention and remotely. What skills and how many years of training were required? Were they cardiologists? anesthetists? urgent?

Author Response

We sincerely appreciate the careful review of our manuscript and the helpful suggestions provided, which have considerably contributed to improving our manuscript. The manuscript has been revised as suggested.

I ask the authors to clarify which were the figures (medical and paramedical) that they managed in the pre-hospital phase, both on site during the intervention and remotely. What skills and how many years of training were required? Were they cardiologists? anesthetists? urgent?

→ Thank you for your pertinent questions. Accordingly, in the Materials and Methods section (page 2, lines 75-82), we have described additionally the scope of work and level of the paramedics at the pre-hospital phase.